# Lighter is Better: Boost Your ViT in Person Re-Identification via Spatial-Aware Token Merging

## Abstract

Vision Transformers (ViTs) have significantly advanced person re-identification (ReID) by providing strong global modeling, but their high computational cost hinders deployment in real-time applications. Existing lightweight ReID methods mostly use token pruning, which can discard discriminative contextual information. Token merging is a moderate alternative, yet existing merging methods target image classification and overlook the local cues that ReID requires. This paper proposes **STM-ReID**, a spatial-aware and training-free token merging framework tailored for ViT-based lightweight ReID. STM-ReID injects information-enhanced spatial awareness into token assessment and uses the resulting scores to guide token matching and fusion, preserving identity-relevant local details while reducing computation. The framework comprises three key components: (i) DSE-Assess, a dynamic spatial-aware entropy weighting for token importance; (ii) CCF-Match, a correlation-guided matching scheme for precise pair selection; (iii) PNR-Fuse, a position response-driven computation strategy for feature aggregation. Extensive experiments on standard ReID benchmarks and general classification datasets show that STM-ReID cuts GFLOPs of the base ViT model by about 24% while keeping accuracy comparable to state-of-the-art methods, yielding a superior accuracy–efficiency trade-off.

## 1 Introduction

Person re-identification (ReID) (Ye et al., 2021; Wu et al., 2023) has emerged as a key problem in computer vision, aiming to retrieve individuals across multiple non-overlapping cameras. This task is widely applied in public security, intelligent surveillance, and criminal investigation, where real-time response and efficient inference are often required. Recent progress in Vision Transformers (ViTs) has brought significant advances to ReID, as their self-attention mechanism enables strong global modeling (He et al., 2021; Wang et al., 2022; Zhu et al., 2022; 2024; Li et al., 2022; 2023). However, ViTs are computationally expensive, and their computational bottleneck primarily resides in feed-forward network modules, whose complexity is closely related to the number of input tokens. Numerous tokens derive from background regions with minimal discriminative contributions or semantically homogeneous foreground areas, leading to high memory usage and slow inference. These limitations make it difficult to deploy ViTs in real-world ReID scenarios, especially on edge devices and real-time surveillance systems.

To reduce this overhead, recent studies on lightweight ReID have mainly adopted token pruning techniques. The key idea is to discard background or redundant tokens that contribute little to discrimination. For example, RCCReID (Wang et al., 2023) introduces a body-masked token selection strategy with an adaptive sliding window to reduce complexity while maintaining accuracy. SUReID (Song & Liu, 2024) further combines hierarchical sparsification with knowledge distillation through parameter-free feature alignment, enabling progressive token reduction. Despite its efficiency, token pruning suffers from the loss of contextual information. By permanently removing tokens, pruning disrupts the spatial and semantic relations among remaining tokens. For ReID, this is especially harmful: identity matching depends not only on global appearance but also on the contextual consistency among local regions. When tokens are discarded, the continuity of such context is broken, making it harder for the model to capture discriminative patterns across complex backgrounds and

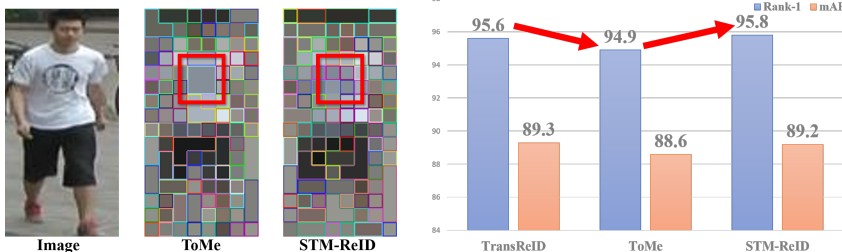

Figure 1: The direct application of generic token compression methods leads to a decrease in ReID accuracy due to the lack of focus on critical local regions.

occluded scenes. This suggests that an ideal compression strategy should not only reduce redundancy but also preserve contextual information across tokens.

A moderate alternative to pruning is token merging, which compresses the sequence by aggregating similar tokens rather than discarding them (Bolya et al., 2023; Feng & Zhang, 2023). Typical merging pipelines operate in three stages: *token assessment* assigns an importance score to each token, *token matching* finds suitable token pairs for fusion, and *token fusion* aggregates the selected tokens and computes their merged representations. Because merging retains relationships among tokens, it better preserves spatial and contextual information and is therefore, in principle, more compatible with ReID, where identity matching depends on both global appearance and local consistency. In this work, we explore bringing token merging into a lightweight ReID design. However, existing merging methods are developed almost exclusively for image classification and show two critical mismatches with ReID needs. First, in the token assessment step, current methods bias importance toward high-level and class-centric signals (e.g., similarity to the class token), ignoring spatial and contextual relations between tokens within the image. Second, token matching and fusion are typically driven exclusively by pair-wise token similarity. While this is adequate for coarse semantic grouping in classification, it can mistakenly merge crucial tokens (e.g., face, clothing logo) with nearby but identity-irrelevant tokens, weakening fine-grained cues. As Figure 1 illustrates, operations that are harmless for classification may remove the very signals ReID requires. These observations motivate the need for spatial-aware assessment and fusion mechanisms tailored to preserve identity-relevant tokens.

Based on the aforementioned concerns, we propose **STM-ReID**, a spatial-aware and training-free token merging framework tailored for ViT-based lightweight ReID. Our approach introduces information-enhanced spatial awareness into token assessment and guides token fusion through these criteria, aiming to preserve discriminative local details while reducing computational cost. Specifically, to overcome the bias of conventional token assessment, we design a Dynamic Spatial-aware Entropy Weighting Assessment (DSE-Assess). It evaluates token importance by integrating spatial layout and local representation concentration, thereby emphasizing identity-relevant regions without relying on class-centric biases. For the matching and fusion stage, we propose a Comprehensive Correlation Function guided Bipartite Soft Matching (CCF-Match) and Position Response Degree Weighted Norm-preserved Token Fusion (PRN-Fuse) that go beyond pure similarity. By considering both semantic affinity and information content, they mitigate the misfusion of discriminative tokens and semantic degradation, maintaining representative features.

Our main contributions can be summarized as follows:

- We propose STM-ReID, a novel spatial-aware lightweight token merging framework for ViT-based ReID. To the best of our knowledge, this is the first work pioneering the adaptation of token merging to ReID.

- We design DSE-Assess, a dynamic spatial-aware entropy-weighted strategy for token assessment, which effectively identifies and emphasizes critical local regions by integrating spatial structure and local information richness, ensuring crucial tokens are accurately evaluated.

- We develop CCF-Match and PRN-Fuse for token matching and fusion, which effectively mitigate the misfusion of discriminative tokens and semantic degradation, therefore maintaining representative features.

- Extensive experiments validate that STM-ReID achieves a significant reduction in computational cost while maintaining competitive accuracy compared to state-of-the-art methods, establishing a superior accuracy-efficiency trade-off for lightweight ReID.

## 2 RELATED WORK

**Token Pruning.** The token pruning strategy compresses the input sequence by removing low-score tokens. For example, DynamicViT (Rao et al., 2021) introduces a lightweight prediction module to estimate token importance scores and dynamically prune low-score tokens at each layer. EViT (Liang et al., 2022) uses a hierarchical approach, classifying tokens into valid and invalid categories based on attention scores, retaining the valid tokens and fusing the invalid ones to reduce redundancy. SPViT (Kong et al., 2022) adopts a stage-wise, layer-by-layer training strategy to progressively prune tokens and optimize model performance.

**Token Merging.** The token merging strategy achieves sequence compression by combining multiple tokens into a single new token. Unlike the pruning strategy, the merging strategy retains more information, thus better maintaining model accuracy. For instance, ToMe (Bolya et al., 2023) and Token Pooling (Marin et al., 2023) methods merge tokens based on semantic similarity, effectively reducing redundant information. BAT (Long et al., 2023) first prioritizes tokens based on their importance, and then selectively retains tokens according to the principle of diversity. These approaches enhance overall efficiency while ensuring the model retains key information.

**Lightweight Person Re-Identification.** Token compression methods in lightweight ReID can be divided into two categories. The first category of methods relies on human structural priors. For instance, RCCReID (Wang et al., 2023) utilizes an edge detection network (Xie & Tu, 2015) to extract image edge maps, and filters background regions by selecting valid tokens, while maintaining recognition accuracy and reducing computational cost. The other category of methods requires no external priors. PAPReID (Ndayishimiye et al., 2025) introduces dynamic token selection, which evaluates importance scores to filter tokens and retains only key tokens for computation. SUReID (Song & Liu, 2024) proposes a hierarchical token sparsification strategy that generates token importance scores, makes differentiable binarization decisions, and dynamically updates masks to progressively eliminate redundant tokens. These methods reduce computational overhead while maintaining accuracy, promoting the development of lightweight ReID.

## 3 METHODOLOGY

In this section, we formally present our STM-ReID, as illustrated in Figure 2. We begin by briefly revisiting TransReID (He et al., 2021), which provides the infrastructure of our approach (Section 3.1). Then, we elaborate on the dynamic spatial-aware entropy weighted token assessment (DSE-Assess) strategy (Section 3.2). Next, we introduce the comprehensive correlation function guided bipartite soft matching (CCF-Match) scheme (Section 3.3). Finally, we propose the position response degree weighted norm-preserved token fusion (PRN-Fuse) paradigm (Section 3.4).

### 3.1 REVISIT TRANSREID

Given a pedestrian image $x \in \mathbb{R}^{H \times W \times C}$, where $H$, $W$, and $C$ represent its height, width, and number of channels, respectively, it is split into $N$ fixed-sized patches. Notably, TransReID employs a sliding window to generate overlapping patches so as to preserve local neighbor structures better. A learnable class token [cls] of dimension $D$, denoted as $z_{\text{cls}} \in \mathbb{R}^D$, is added at the beginning of the input sequence. Position embeddings and side information embeddings are then added to form the token sequence $Z_0 \in \mathbb{R}^{(N+1) \times D}$, which is passed through $L$ transformer layers.

Within each transformer layer, the computational process can be formulated as follows:

$$Z'_l = \text{MHSA}(\text{LN}(Z_{l-1})) + Z_{l-1}, \quad Z_l = \text{FFN}(\text{LN}(Z'_l)) + Z'_l, \quad l \in [1, L], \tag{1}$$

where MHSA$(\cdot)$ represents multi-head self-attention (Vaswani et al., 2017), FFN$(\cdot)$ denotes feed-forward network, and LN is layer normalization. Let $Q, K, V \in \mathbb{R}^{(N+1) \times D}$ denote the query, key,

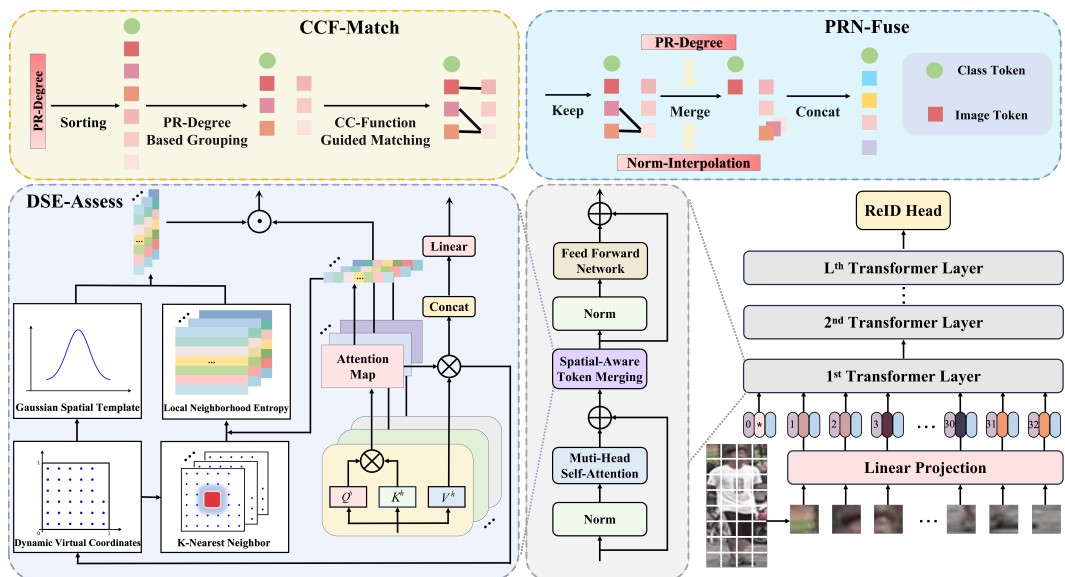

Figure 2: The structure of STM-ReID. It is mainly composed of the DSE-Assess strategy, the CCF-Match scheme, and the PRN-Fuse paradigm.

and value matrices, MHSA performs parallel attention computation on the input:

$$\text{Attention}(Q, K, V) = A \times V, \quad A = \text{Softmax}\left(\frac{QK^T}{\sqrt{D_h}}\right), \tag{2}$$

where $D_h = D/H$ denotes the output dimension of a single head among $H$ attention heads. Similarly, the attention scores between the class token $z_{\text{cls}}$ and other tokens can be denoted as:

$$A(z_{\text{cls}}, :) = \text{Softmax}\left(\frac{Q_{\text{cls}}K^T}{\sqrt{D_h}}\right). \tag{3}$$

Current token assessment approaches typically average outputs across all attention heads, while neglecting spatial and contextual relations between tokens. This may lead to the underestimation of discriminative tokens in critical local regions, thereby losing crucial retrieval cues.

### 3.2 DYNAMIC SPATIAL-AWARE ENTROPY WEIGHTED TOKEN ASSESSMENT

To mitigate information loss caused by naive head averaging, we propose **DSE-Assess**, which computes an entropy measure for each spatial location of the input image and uses local neighborhood entropy to differentiate the importance of each attention head dynamically. Attention heads that exhibit lower entropy, which indicate more concentrated distributions, receive higher weights. This approach relies only on MHSA outputs and requires no extra supervision, enabling a **training-free** assessment mechanism (more details are in the **Appendix**).

TransReID uses the sliding window to generate overlapping patches, and subsequent merging may change the number and arrangement of tokens across layers. Traditional approaches based on fixed dimensions or positions are rendered inapplicable, as they cannot predetermine the variable token configurations (quantity and arrangement) at each layer. To establish spatial location relationships, we introduce *Dynamic Virtual Coordinates* (DVC) that map an arbitrary number of tokens onto a virtual grid. Given $N$ image tokens at the current layer (excluding the class token), we arrange them sequentially into a virtual $H_{\text{v}} \times W_{\text{v}}$ grid with $H_{\text{v}} = \lceil\sqrt{N}\rceil$, $W_{\text{v}} = \lceil N/H_{\text{v}}\rceil$, and index tokens by $i \in \{0, \ldots, N-1\}$. The normalized coordinates of token $i$ are $x_i = \frac{\lfloor i/W_{\text{v}}\rfloor + 0.5}{H_{\text{v}}}$, $y_i = \frac{i\%W_{\text{v}} + 0.5}{W_{\text{v}}}$, so that $(x_i, y_i) \in (0, 1)^2$ and token layout reflects the virtual spatial arrangement.

In ReID datasets, pedestrian crops are typically centered in the image. To encode this prior, we define *Adaptive Gaussian Spatial Template* (AdaptGST) that emphasizes central regions. For a layer with

$N$ tokens, we set the Gaussian scale $\sigma = \alpha\sqrt{N}$ (we use $\alpha = 0.1$), and the spatial prior for token $i$ is

$$\mathbf{W}_{\text{space}}(x_i, y_i) = \exp\left(-\frac{(x_i - 0.5)^2 + (y_i - 0.5)^2}{2\sigma^2}\right). \tag{4}$$

This adaptive template adaptively adjusts the central prior intensity according to the token density at the current layer.

To eliminate dependency on fixed grids and flexibly perceive spatial and contextual relations, we form for each token $z_i$ a neighborhood $\mathcal{N}(i)$ consisting of its $k$ nearest tokens under Euclidean distance in the virtual coordinates $(x_i, y_i)$. Let $\mathbf{A}^{(h)}(z_{\text{cls}}, z_j)$ denote the attention weight from the class token $z_{\text{cls}}$ to token $z_j$ in head $h$. The local neighborhood entropy for head $h$ at token $i$ is

$$\mathcal{H}^{(h)}(z_i) = -\sum_{j \in \mathcal{N}(i)} \mathbf{A}^{(h)}(z_{\text{cls}}, z_j) \cdot \log \mathbf{A}^{(h)}(z_{\text{cls}}, z_j), \tag{5}$$

A lower entropy indicates a more concentrated attention distribution of head $h$ on the corresponding neighborhood. Subsequently, we integrate the local neighborhood entropy with spatial weights to derive the attention weight of head $h$ for token $z_i$:

$$\mathcal{R}^{(h)}(z_i) = \frac{\mathbf{W}_{\text{space}}(z_i) \cdot \exp\left(-\mathcal{H}^{(h)}(z_i)\right)}{\sum_{h'=1}^{H} \mathbf{W}_{\text{space}}(z_i) \cdot \exp\left(-\mathcal{H}^{(h')}(z_i)\right)}. \tag{6}$$

The proposed formulation dynamically allocates the attention weights of each head by integrating spatial priors with local neighborhood entropy through a normalized exponential function. Compared to conventional averaging approaches, our method amplifies contributions from critical heads (with low entropy and high spatial weights), thereby achieving enhanced focus on crucial details.

Ultimately, the importance of token $z_i$ is defined as the weighted summation of attention weights across all heads with the spatial prior. We call this the *Position Response Degree* (PR-Degree), which is used to guide subsequent token merging:

$$\mathcal{S}(z_i) = \sum_{h=1}^{H} \mathcal{R}^{(h)}(z_i) \cdot \mathbf{A}^{(h)}(z_{\text{cls}}, z_j), \quad i \in [0, N]. \tag{7}$$

### 3.3 COMPREHENSIVE CORRELATION FUNCTION GUIDED BIPARTITE SOFT MATCHING

**PR-Degree Based Grouping.** After computing the PR-Degree for each image token, we keep the class token $z_{\text{cls}}$ separate and consider the set of image tokens indexed by $\mathcal{I} = \{1, \ldots, N\}$. Let $\mathcal{S} = \{\mathcal{S}(z_i)\}_{i \in \mathcal{I}}$ be the PR-Degree values for these tokens. We form a descending permutation $\text{Index} = (p_1, \ldots, p_N)$ by sorting $\mathcal{S}$ in decreasing order, i.e. $\mathcal{S}(z_{p_1}) \geq \mathcal{S}(z_{p_2}) \geq \cdots \geq \mathcal{S}(z_{p_N})$. We then split this ordered list into two subsequences by alternating positions: $\text{Index}_A = \{0, p_1, p_3, p_5, \ldots\}$, $\text{Index}_B = \{p_2, p_4, p_6, \ldots\}$, where the class token index 0 is explicitly inserted into $\text{Index}_A$ to help preserve core features during matching. Based on this partitioning, we extract the corresponding tokens from $Z \in \mathbb{R}^{(N+1) \times D}$ and form two subsequences $Z_A = Z[\text{Index}_A]$ and $Z_B = Z[\text{Index}_B]$. The traditional fixed odd-even grouping may lead to semantically similar or redundant tokens being clustered within the same group, thereby hindering cross-group pairing (as matching occurs exclusively between groups). Our PR-Degree metric reflects task-specific priority of tokens. This priority-ordered grouping strategy effectively disrupts fixed token distributions and enhances semantically correlated merging.

**Comprehensive Correlation Function.** Given $Z \in \mathbb{R}^{(N+1) \times D}$, the goal at each layer is to merge $r$ pairs of redundant tokens and produce a compressed sequence $\hat{Z} \in \mathbb{R}^{(N+1-r) \times D}$. To guide matching while protecting identity-relevant crucial tokens, we define a *Comprehensive Correlation Function* (CC-Function) that combines feature similarity with token responsiveness to evaluate inter-group token relationships.

Since the key matrix has distilled critical information of each token through self-attention, we use the key vectors (rows of the key matrix) as compact semantic descriptors for feature similarity. For tokens $z_i$ and $z_j$ with key vectors $k_i, k_j \in \mathbb{R}^D$, we define a cosine-based similarity:

$$F_{\text{sim}}(k_i, k_j) = \frac{1}{2}\left(\frac{k_i \cdot k_j^T}{\| k_i \|_2 \cdot \| k_j \|_2} + 1\right), \tag{8}$$

where $||\cdot||_2$ represents the L2-norm. The final results are normalized and shifted to the [0,1] interval.

However, relying solely on inter-token similarity as the merging criterion entails potential risks. Semantically similar tokens in feature space are not necessarily redundant, particularly when they are located in critical local regions essential for identity modeling. Erroneous merging of such tokens may lead to irreversible loss of discriminative details, ultimately degrading model performance. Therefore, we design the responsiveness metric as an auxiliary matching criterion to evaluate each token's contribution to the global identity representation, thereby determining which tokens should be prioritized for retention. The aforementioned PR-Degree is leveraged to formulate the responsiveness scoring function $F_{\text{res}}$. Specifically, a higher PR-Degree signifies that the token possesses stronger semantics for identity discrimination, thus should be preserved from merging so as to mitigate the risk of losing crucial details. Conversely, a token with a lower PR-Degree indicates diminished global influence and is likely a semantically redundant token. The responsiveness scoring function between token pairs is expressed as follows:

$$F_{\text{res}}(z_i, z_j) = \hat{\mathcal{S}}(z_i)\cdot \hat{\mathcal{S}}(z_j), \quad \hat{\mathcal{S}}(z_i) = \frac{1/\mathcal{S}(z_i)}{\max_k\left(1/\mathcal{S}(z_k)\right)}, \quad k \in [1, N]. \tag{9}$$

When both tokens exhibit comparatively high PR-Degree, their responsiveness score remains lower, thereby effectively preventing the misintegration of crucial tokens. The final CC-Function value is denoted as the product of the similarity and responsiveness with tunable exponents:

$$F(z_i, z_j) = \left(F_{\text{sim}}(k_i, k_j)\right)^{\delta_1} \cdot \left(F_{\text{res}}(z_i, z_j)\right)^{\delta_2}, \tag{10}$$

where $\delta_1, \delta_2 > 0$ control the relative emphasis on semantic similarity and responsiveness. A higher CC-Function value indicates a greater probability of token fusion between the corresponding pair.

**Bipartite Soft Matching with CC-Function.** We perform bipartite soft matching (Bolya et al., 2023) between tokens in $Z_A$ and $Z_B$ using the CC-Function. First, we compute the matrix of correlation scores $F(z_i, z_j)$ for all $z_i \in Z_A$ and $z_j \in Z_B$. Then, we identify the candidate pairs by selecting the highest-scoring matches. Finally, all candidate pairs are sorted in descending order of scores, and the top-$r$ highest-scoring token pairs are selected as the ultimate fusion targets, which are then passed to the fusion module (PRN-Fuse). For more details of the algorithm, please refer to the **Appendix**.

### 3.4 POSITION RESPONSE DEGREE WEIGHTED NORM-PRESERVED TOKEN FUSION

Traditional token-size-based fusion commonly uses weighted averaging to obtain a fused token. Such averaging, however, ignores differences in token importance: semantically important tokens can be overly smoothed by redundant ones, degrading feature discriminability.

To this end, we introduce *Position Response Degree Weighted Norm-Preserved Token Fusion* (PRN-Fuse), which incorporates PR-Degree into the weighting strategies to mitigate the shortcomings of token-size-based methods. For $T$ tokens to be merged, the fused feature representation is defined as:

$$z_{\text{avg}} = \sum_{i=1}^{T} w_i z_i, \quad w_i = \frac{\mathcal{S}(z_i)}{\sum_{k=1}^{T} \mathcal{S}(z_k)}, \quad \sum_{i=1}^{T} w_i = 1. \tag{11}$$

The weighted summation is essentially a linear combination. According to the norm inequality $\|z_{\text{avg}}\| = \left\|\sum_{i=1}^{T} w_i z_i\right\| \le \max \|z_i\| \ (i \in [1, T])$, the fused norm is inevitably bounded by the maximum norm of the original tokens. Notably, as the network depth increases, this norm diminishment becomes more pronounced, resulting in the weakened expressiveness of crucial tokens (more analysis can be found in the **Appendix**). Therefore, we employ max-norm linear interpolation for token merging. Building upon PR-Degree weighted fusion, we compute the $L2$-norm of the fused feature and perform normalization. By selecting the maximum $L2$-norm from the original features of each token pair, we apply scaling to the normalized fused feature, thereby restoring the scale information of the feature representation:

$$\hat{z}_{\text{avg}} = \frac{z_{\text{avg}}}{\|z_{\text{avg}}\|_2} \times \max_{1 \le i \le T} \|z_i\|_2. \tag{12}$$

Table 1: Comparison of GFLOPs, inference time (s), throughput (img/s), Rank-1 (%), and mAP (%) with state-of-the-art token compression methods on MSMT17 and Market1501.

| Methods | GFLOPs ↓ | Inference ↓ | Throughput ↑ | MSMT17 | | Market1501 | |
|---|---|---|---|---|---|---|---|
| | | | | Rank-1 ↑ | mAP ↑ | Rank-1 ↑ | mAP ↑ |
| Specific Compression Methods for ReID | | | | | | | |
| RCCReID (Wang et al., 2023) | 17.18 | - | - | 82.2 | 63.0 | 94.9 | 88.0 |
| SUReID (Song & Liu, 2024) | - | - | - | - | - | 94.5 | 87.2 |
| PAPReID (Ndayishimiye et al., 2025) | 16.86 | - | - | - | - | 79.1 | 77.3 |
| Generic Compression Methods for ViTs | | | | | | | |
| DPC-KNN (Du et al., 2016) | 17.24 | 54.79 | 351.89 | 84.5 | 65.7 | 94.7 | 87.9 |
| DynamicViT (Rao et al., 2021) | 19.30 | 58.31 | 330.68 | 84.5 | 66.6 | 94.7 | 87.5 |
| EViT (Liang et al., 2022) | 17.50 | 54.27 | 355.27 | 84.9 | 67.0 | 94.8 | 88.0 |
| ATS (Fayyaz et al., 2022) | 17.39 | 55.34 | 348.43 | 81.2 | 58.8 | 93.4 | 84.7 |
| ToMe (Bolya et al., 2023) | **15.53** | **45.99** | **419.25** | 85.0 | 67.1 | 94.9 | 88.6 |
| TransReID (He et al., 2021) | 20.41 | 57.43 | 335.73 | **85.3** | **68.1** | 95.6 | **89.3** |
| **STM-ReID (Ours)** | **15.53** $\downarrow^{23.9\%}$ | 46.28 | 416.62 | **85.3** | 67.8 | **95.8** $\uparrow^{0.2\%}$ | 89.2 |

Table 2: Comparison of GFLOPs and Rank-1 (%) with ToMe on Market1501 when STM-ReID merges $r$=17 tokens per layer while ToMe employs different merging numbers $r$.

|  (a) $r$=17 | | | (b) $r$=8 | | | (c) $r$=6 | | |
|---|---|---|---|---|---|---|---|---|
| Methods | GFLOPs | Rank-1 | Methods | GFLOPs | Rank-1 | Methods | GFLOPs | Rank-1 |
| ToMe | **10.31** | 94.4 | ToMe | 15.53 | 94.9 | ToMe | 16.69 | 95.1 |
| STM-ReID | **10.31** | **95.2** | STM-ReID | **10.31** | **95.2** | STM-ReID | **10.31** | **95.2** |
| *Comparison* | ↓ 0% | ↑ 0.8% | *Comparison* | ↓ 33.6% | ↑ 0.3% | *Comparison* | ↓ 38.2% | ↑ 0.1% |

# 4 EXPERIMENTS

## 4.1 DATASETS AND EVALUATION METRICS

**Datasets.** We evaluate our method on two public ReID benchmarks: Market1501 (Zheng et al., 2015) and MSMT17 (Wei et al., 2018). The Market1501 contains 32,668 images of 1,501 identities acquired from 6 surveillance cameras, while the MSMT17 consists of 126,441 images of 4,101 identities captured by 15 cameras. To further validate the generalization ability of our proposed method on the image classification task, we also conduct comparative experiments between STM-ReID and token compression methods (Ndayishimiye et al., 2025; Wang et al., 2023; Song & Liu, 2024; Liang et al., 2022; Fayyaz et al., 2022; Du et al., 2016; Rao et al., 2021; Bolya et al., 2023) on the ImageNet-1k (Deng et al., 2009) dataset.

**Evaluation Metrics.** We employ Giga Floating-point Operations (GFLOPs) as the metric to quantify the computational complexity of the model. Besides, we also employ inference time and throughput to evaluate the computational efficiency, which refer to the time required for the model to complete inference on a dataset and the number of images processed per second respectively. Furthermore, we utilize the cumulative matching characteristic (CMC) curve and mean average precision (mAP) as metrics to evaluate the retrieval accuracy. Rank-$k$ in the CMC curve measures the probability of finding a correct match within the top-$k$ results, while mAP evaluates average retrieval performance across all queries.

## 4.2 IMPLEMENTATION DETAILS

The proposed STM-ReID is implemented in PyTorch following TransReID, with all experiments conducted on a single A100 GPU. We employ TransReID as our backbone network, where the image encoder is initialized with ViT-B/16 on ReID datasets. On ImageNet-1k, we adopt three different backbone networks for fair comparison with other methods, namely DeiT-T, DeiT-S, and DeiT-B. All images are resized to 256×128. The batch size is set to 64, with each mini-batch containing 16 identities and 4 images per identity. We select $k$=9 nearest tokens as the neighborhood. Unless otherwise specified, our token merging is applied to layers 1 to 10 of the model, with the number

Table 3: Comparison of Rank-1 (%) and mAP (%) with state-of-the-art ReID methods on Market1501 and MSMT17. **Note that STM-ReID compresses approximately 24% of the tokens, while other methods *DO NOT* perform compression here.**

| Backbone | Methods | Market1501 | | MSMT17 | |
|---|---|---|---|---|---|
| | | Rank-1 ↑ | mAP ↑ | Rank-1 ↑ | mAP ↑ |
| CNN | DRL-Net (Jia et al., 2022) | 94.7 | 86.9 | 78.4 | 55.3 |
| | AGW (Ye et al., 2022) | 95.1 | 87.8 | 68.3 | 49.3 |
| | MSINet (Gu et al., 2023) | 95.3 | 89.6 | 81.0 | 59.6 |
| | HashReID (Nikhal et al., 2024) | 94.2 | 84.9 | 76.8 | 51.4 |
| | DCR (Yang & Xu, 2025) | 95.3 | 88.3 | 81.0 | 56.4 |
| ViT | PFD (Wang et al., 2022) | 95.5 | 89.7 | - | - |
| | DCAL (Zhu et al., 2022) | 94.7 | 87.5 | 83.1 | 64.0 |
| | AAformer (Zhu et al., 2024) | 95.4 | 87.7 | 83.6 | 63.2 |
| | ProFD (Cui et al., 2024) | 95.1 | 90.0 | - | - |
| | MV-3DSReID (Yu et al., 2024) | 95.7 | **90.2** | - | - |
| | TransReID (He et al., 2021) | 95.6 | 89.3 | **85.3** | **68.1** |
| | **STM-ReID (Ours)** | **95.8** | 89.2 | **85.3** | 67.8 |

Table 4: Comparison of Rank-1 (%) with state-of-the-art token compression methods using multiple backbones on ImageNet-1k.

| Methods | DeiT-S | DeiT-T | DeiT-B |
|---|---|---|---|
| DPC-KNN (Du et al., 2016) | 78.85 | 71.10 | 79.06 |
| DynamicViT (Rao et al., 2021) | 79.17 | 67.40 | 80.68 |
| EViT (Liang et al., 2022) | 79.30 | 71.06 | 80.99 |
| ATS (Fayyaz et al., 2022) | 79.09 | 70.77 | 80.78 |
| ToMe (Bolya et al., 2023) | 79.39 | 71.10 | 81.05 |
| **STM-ReID (Ours)** | **79.41** | **71.43** | **81.32** |

of merged tokens per layer set to $r=8$. Please refer to the **Appendix** for the determination of the parameters $k$ and $r$.

## 4.3 COMPARISON WITH STATE-OF-THE-ART METHODS

To verify the effectiveness of the proposed method, we first conduct experiments on computational complexity and retrieval accuracy using the MSMT17 and Market1501 datasets. We comprehensively compare STM-ReID with previous state-of-the-art token compression methods, and all results are reported without re-ranking. For fair comparison, we adopt the same settings as ToMe, i.e., merging $r=8$ tokens per layer, resulting in the same GFLOPs. As shown in Table 1, STM-ReID's inference efficiency is slightly lower than that of ToMe. This is because ToMe does not evaluate the semantic importance of tokens and instead directly performs grouping and merging operations. In contrast, we first employ DSE-Assess to ensure crucial tokens are retained, thus outperforming ToMe significantly across all four accuracy metrics on both datasets, yet the increased inference overhead is almost negligible. Compared to TransReID, STM-ReID reduces GFLOPs by 23.9%, shortens inference time by 19.4%, and increases throughput by 24.1%, while achieving comparable accuracy. This indicates that STM-ReID establishes a favorable balance between recognition accuracy and computational efficiency for lightweight ReID.

For an intuitive comparison with ToMe, we fix $r=17$ for STM-ReID and examine the differences when ToMe adopts different $r$. As illustrated in Table 2, when STM-ReID merges 17 tokens per layer, its GFLOPs is 10.31 (half of TransReID) and Rank-1 is 95.2%. (a) When ToMe also uses $r=17$, STM-ReID exhibits significant performance advantages. (b) When ToMe chooses $r=8$, STM-ReID's GFLOPs is only 66.4% of ToMe's, while maintaining a certain performance advantage. (c) When ToMe adopts $r=6$ to maximize accuracy as much as possible, STM-ReID's GFLOPs is only 61.8% of ToMe's, while its performance still remains superior to ToMe's. The above results fully demonstrate that, in terms of lightweight ReID task, STM-ReID significantly outperforms the baseline ToMe across multiple dimensions, thanks to the innovative designs of DSE-Assess, CCF-Match, and PRN-Fuse. These results fully demonstrate the superiority of STM-ReID.

Table 5: Ablation studies on Market1501 to evaluate the effectiveness of proposed components.

| Index | Components | | | | Market1501 | |
|---|---|---|---|---|---|---|
| | Baseline | DSE-Assess | CCF-Match | PRN-Fuse | Rank-1 ↑ | mAP ↑ |
| 1 | ✓ | - | - | - | 94.9 | 88.6 |
| 2 | ✓ | ✓ | ✓ | - | 95.4 | 88.9 |
| 3 | ✓ | ✓ | - | ✓ | 95.5 | 89.1 |
| 4 | ✓ | ✓ | ✓ | ✓ | **95.8** | **89.2** |

Table 6: The impact of temperature coefficients on model performance on MSMT17.

| (a) Temperature coefficient - $\delta_1$ | | | | | | (b) Temperature coefficient - $\delta_2$ | | | | | |
|---|---|---|---|---|---|---|---|---|---|---|---|
| $\delta_1$ | 1 | 1/5 | 1/10 | 1/20 | 1/40 | $\delta_2$ | 1 | 1/5 | 1/10 | 1/20 | 1/40 |
| Rank-1 | **83.0** | 81.1 | 78.2 | 74.6 | 69.8 | Rank-1 | 69.7 | 79.1 | 81.9 | **83.0** | 82.9 |
| mAP | **64.3** | 61.2 | 57.6 | 52.5 | 47.4 | mAP | 47.4 | 58.8 | 62.0 | **64.3** | 63.7 |

To more comprehensively validate the advantages of STM-ReID, we perform comparative experiments between STM-ReID and state-of-the-art ReID methods on Market1501 and MSMT17. STM-ReID uses the default setting ($r=8$) and achieves approximately 24% token compression compared to TransReID, while other methods do not perform compression. As shown in Table 3, STM-ReID achieves competitive performance on both datasets. This is attributed to the fact that DSE-Assess fully considers the semantic importance of tokens, thus retaining the crucial retrieval cues for ReID.

To demonstrate the generalization ability of STM-ReID, we also conduct experiments on the ImageNet-1k dataset. Table 4 compares the Rank-1 accuracy of various state-of-the-art token compression methods. When using the same backbone networks as other methods (i.e., DeiT-S, DeiT-T, and DeiT-B), STM-ReID achieves the highest accuracy in all cases. This indicates that our method is not only applicable to the ReID task but also able to achieve favorable performance in the classification task.

### 4.4 ABLATION STUDY

To evaluate the effectiveness of key components in our STM-ReID, we conduct a series of ablation studies on the Market1501 dataset. ToMe is adopted as the baseline method. The evaluated components include the DSE-Assess strategy, the CCF-Match scheme, and the PRN-Fuse paradigm, with results summarized in Table 5. Progressive component additions yield consistent metric gains, validating each module's contribution to feature representation and retrieval accuracy. The full model (Index 4) achieves optimal performance, demonstrating the cumulative effectiveness of integrating DSE-Assess, CCF-Match, and PRN-Fuse.

The temperature coefficients are employed to balance the contributions of similarity and responsiveness in the CC-Function guided bipartite soft matching process. Therefore, to evaluate the impact of temperature coefficients on model performance, we also perform comparative experiments on the MSMT17 dataset using two distinct settings. As can be seen from Table 6, as $\delta_1$ increases, the model performance gradually decreases; whereas for $\delta_2$, it exhibits a trend of first increasing and then decreasing. The model performance reaches its best when adopting $\delta_1 = 1$ and $\delta_2 = 1/20$.

### 5 CONCLUSION

This paper presents STM-ReID, a novel training-free spatial-aware token merging paradigm for ViT-based lightweight ReID. It achieves robust performance through three core designs: a DSE-Assess strategy, a CCF-Match scheme, and a PRN-Fuse paradigm. Through the DSE-Assess strategy, we successfully highlight critical local regions and allocate higher weights to crucial tokens. Through the CCF-Match scheme and PRN-Fuse paradigm, we effectively avoid misintegration of crucial tokens and alleviate the norm diminishment. Our method achieves a 24% GFLOPs reduction in the base model while maintaining competitive accuracy with state-of-the-art methods.

## 6 ETHICS STATEMENT

This research strictly adheres to the ICLR Code of Ethics. No human or animal subjects were involved, and datasets (Market1501, MSMT17, ImageNet-1k) were sourced per usage guidelines to avoid privacy infringement. Deliberate measures were taken to eliminate biases and discriminatory outcomes, with no personally identifiable information used and no privacy/security risks posed. We uphold transparency and academic integrity throughout.

## 7 REPRODUCIBILITY STATEMENT

Every effort ensures the reproducibility of reported results. Code will be publicly available post-acceptance to support replication. The paper details the experimental setup (training procedures, model configurations, hardware specs) and comprehensive implementation details. All datasets are publicly available, guaranteeing consistent evaluation. These measures enable fellow researchers to replicate our work and advance the field.

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

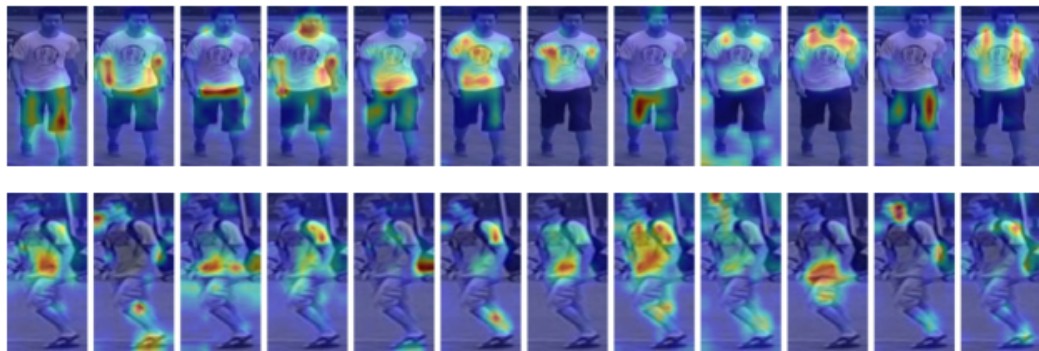

Figure 3: Spatial distribution of attentional foci across distinct attention heads. Each row corresponds to an individual image sample, with the left-to-right sequence demonstrating the 12 attention heads in the fourth transformer layer of TransReID.

## A ANALYSIS

### A.1 RATIONALE FOR DIFFERENTIAL WEIGHTING ACROSS ATTENTION HEADS

Existing token assessment methods typically compute either similarity scores with the class token or the sum of attention weights allocated by other query tokens when serving as a key. Then they average the outputs of these attention heads as the importance of the current token. However, individual heads usually exhibit differential responses towards different patterns. The simple averaging operation causes the loss of this diversity, leading to systemic underestimation of crucial tokens located in critical local regions.

To illustrate this issue, we visualize the spatial distributions of attentional foci across distinct attention heads. As can be seen in Figure 3, it demonstrates divergent spatial foci across heads: selective focus on foreground anatomical regions (e.g., head and limbs), appendages (e.g., backpack), and incidental background elements. This visualization conclusively validates the suboptimal nature of direct averaging across attention heads. Therefore, we propose the dynamic spatial-aware entropy weighted token assessment (DSE-Assess) strategy to assign higher importance to tokens in critical local regions beneficial for identity discrimination.

### A.2 RATIONALE FOR UTILIZING PR-DEGREE DURING TOKEN MERGING.

Traditional token size-based methods adopt weighted averaging strategies to accomplish token merging. We point out that these strategies induce feature attenuation by neglecting contribution disparity among different tokens in identity matching. This problem becomes particularly pronounced when merging multiple tokens simultaneously.

To substantiate this claim, Figure 4 quantitatively illustrates the inverse correlation between average cosine similarity and token merging quantity. It demonstrates that the average cosine similarity progressively declines as the number of tokens to be merged increases. This downward trend indicates that merging an excessive number of tokens in a single operation may force the fusion of semantically divergent tokens, thereby obscuring the salient expression of critical information. Therefore, we incorporate PR-Degree into the weighting strategy to mitigate the shortcomings of token size-based methods and preserve feature discriminability.

### A.3 RATIONALITY ANALYSIS OF DSE-ASSESS

As formalized in Equation (7), our token assessment strategy (DSE-Assess) employs PR-Degree to quantify the importance of each token. This dual-gating mechanism requires both high class token attention score and attention head weight to assign significance to a token. Even if some attention heads exhibit high activations on redundant regions in terms of class token attention scores, their low weighting coefficients (derived from high local neighborhood entropy and/or low spatial weights)

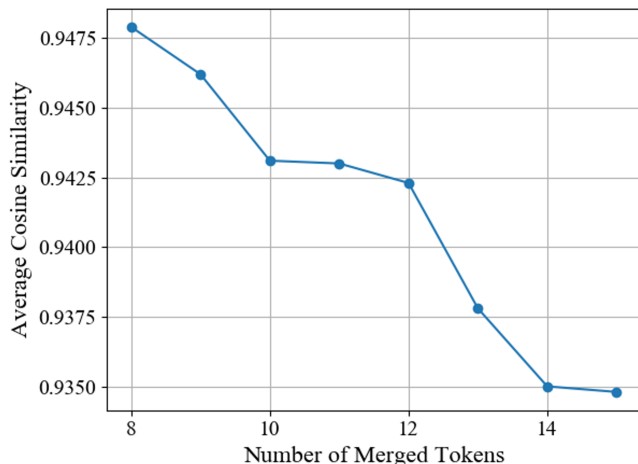

Figure 4: The relationship between average cosine similarity and token merging quantity.

inherently suppress such tokens during PR-Degree computation. This composite mechanism relies solely on the outputs of multi-head self-attention and requires no additional supervision, yet effectively identifies identity-discriminative tokens to enhance the retrieval ability of the ReID model.

### A.4 EXPLANATION OF TRAINING-FREE PROPERTY

In this paper, training-free means all mechanisms of STM-ReID are implemented based on pre-trained models, without introducing new learnable parameters or requiring additional training, fine-tuning, or distillation of the models. Token merging via designed strategies is completed only during the inference phase. The implementation of DSE-Assess, CCF-Match, and PRN-Fuse all rely on existing attention weights (e.g., outputs of multi-head self-attention) and feature vectors (e.g., K vectors) of the pre-trained models. All operations are parameter-free computations, involving no training updates. This design allows STM-ReID to be directly deployed on pre-trained models, avoiding extra training costs, and is particularly suitable for resource-constrained real-world scenarios.

## B ALGORITHMS

The details of our proposed CCF-Match scheme and PRN-Fuse paradigm (with a regular merge operation) are described in Algorithms 1 2 3.

## C EXPERIMENTS

### C.1 THE RELATIONSHIP BETWEEN GFLOPS AND TOKEN MERGING QUANTITY

Our experiments demonstrate the quantitative relationship between the number of merged tokens per layer ($r$) and GFLOPs (with merging applied only to the first 10 layers to preserve performance), as shown in Figure 5. Increasing $r$ progressively reduces GFLOPs, exhibiting an approximately linear correlation. This confirms that the token merging quantity directly dictates computational complexity. Such parametric control enables effective trade-offs between model performance and computational costs, providing theoretical foundations for optimizing inference efficiency across deployment scenarios.

### C.2 TEST OF THE NUMBER OF NEAREST TOKENS WITHIN A NEIGHBORHOOD

We investigate the optimal number of nearest neighbor tokens $k$ within a neighborhood, as detailed in Table 7. Both Rank-1 and mAP initially increase with larger $k$ values before declining, achieving optimal accuracy at $k=9$. Therefore, we utilize $k=9$ across all experimental configurations.

---

**Algorithm 1** CCF-Match

---

**Input**: The Key $K$, PR-Degree $\mathcal{S}$, number of merged tokens $r$
**Output**: Indices of unmatched token pairs $T_{ui}$, reserved indices of matched token pairs $T_{si}$ and $T_{di}$

1: $sortScore \leftarrow \text{argsort}(\mathcal{S}, order = descending)$
2: $Index_A \leftarrow sortScore_{even}$
3: $Index_B \leftarrow sortScore_{odd}$
4: $\hat{\mathcal{S}} \leftarrow \frac{1/\mathcal{S}}{\max_k(1/\mathcal{S}_k)}$
5: $K_A \leftarrow K[Index_A], K_B \leftarrow K[Index_B]$
6: $\mathcal{S}_A \leftarrow \hat{\mathcal{S}}[Index_A]), \mathcal{S}_B \leftarrow \hat{\mathcal{S}}[Index_B])$
7: $F_{sim} \leftarrow \frac{1}{2}\left(\frac{K_A \cdot K_B^T}{\|K_A\|_2 \cdot \|K_B\|_2} + 1\right)$
8: $F_{res} \leftarrow \mathcal{S}_A \times \mathcal{S}_B^T$
9: $Score \leftarrow F_{sim} \cdot F_{res}$
10: $M_v \leftarrow \max(Score, dim = 1)$
11: $M_i \leftarrow \text{argmax}(Score, dim = 1)$
12: $M_{vs} \leftarrow \text{argsort}(M_v, order = descending)$
13: $T_{si} \leftarrow M_{vs}[: r]$
14: $T_{ui} \leftarrow M_{vs}[r :]$

---

**Algorithm 2** Merge

---

**Input**: Token sequence $T$, indices of unmatched token pairs $T_{ui}$, reserved indices of matched token pairs $T_{si}$ and $T_{di}$, aggregation operation $mode$
**Output**: Merged token sequence $T_m$

1: $T_{even} \leftarrow T[Index_A], T_{odd} \leftarrow T[Index_B]$
2: $T_u \leftarrow T_{even}[T_{ui}]$
3: $T_s \leftarrow T_{even}[T_{si}]$
4: $T_d \leftarrow T_{odd} \cdot \text{scatter\_reduce}(T_{di}, T_s, mode)$
5: $T_m \leftarrow \text{concatenate}(T_{di}, T_s, mode)$

---

**Algorithm 3** PRN-Fuse

---

**Input**: Token sequence $Z$, PR-Degree $\mathcal{S}$, size of token sequence $Size$
**Output**: Merged token sequence $Z_m$

1: $Z_{score} \leftarrow Merge(Z \times \mathcal{S}, mode = \text{"sum"})$
2: $\mathcal{S} \leftarrow Merge(\mathcal{S}, mode = \text{"sum"})$
3: $Size \leftarrow Merge(Size, mode = \text{"sum"})$
4: $Z_{score} \leftarrow Z_{score}/\mathcal{S}$
5: $Z_{norm} \leftarrow \|Z\|_2$
6: $Z_{score\_norm} \leftarrow \|Z_{score}\|_2$
7: $Z_{norm\_max} \leftarrow Merge(Z_{norm}, mode = \text{"amax"})$
8: $Z_m \leftarrow \frac{Z_{score}}{Z_{score\_norm}} \times Z_{norm\_max}$

---

Table 7: Rank-1 (%) and mAP (%) under varying $k$ values.

| $k$ | 1 | 4 | 9 | 16 | 25 |
|---|---|---|---|---|---|
| Rank-1 | 95.4 | 95.5 | 95.8 | 95.5 | 95.4 |
| mAP | 89.1 | 89.1 | 89.2 | 89.1 | 89.1 |

### C.3 COMPARISON WITH ToMe UNDER VARYING GFLOPs ON MARKET1501

We conduct comparative experiments with ToMe on the Market1501 dataset to analyze performance disparities under varying GFLOPs. As illustrated in Figure 6, STM-ReID demonstrates consistent

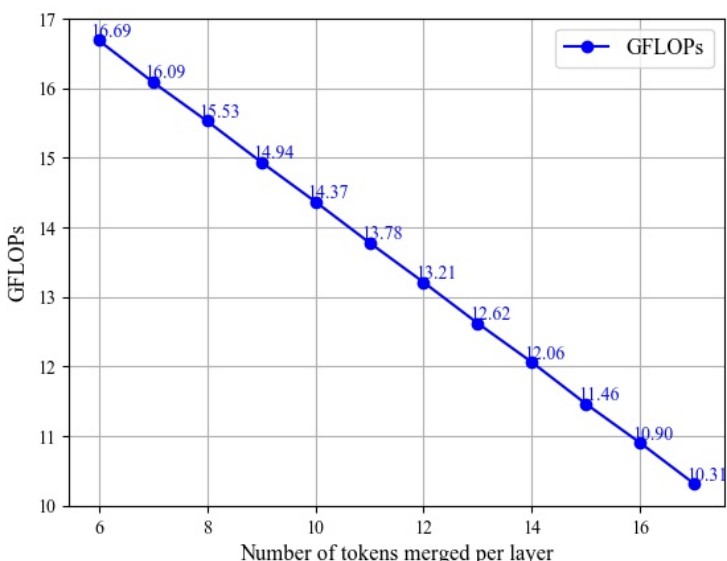

Figure 5: The relationship between GFLOPs and number of merged tokens $r$.

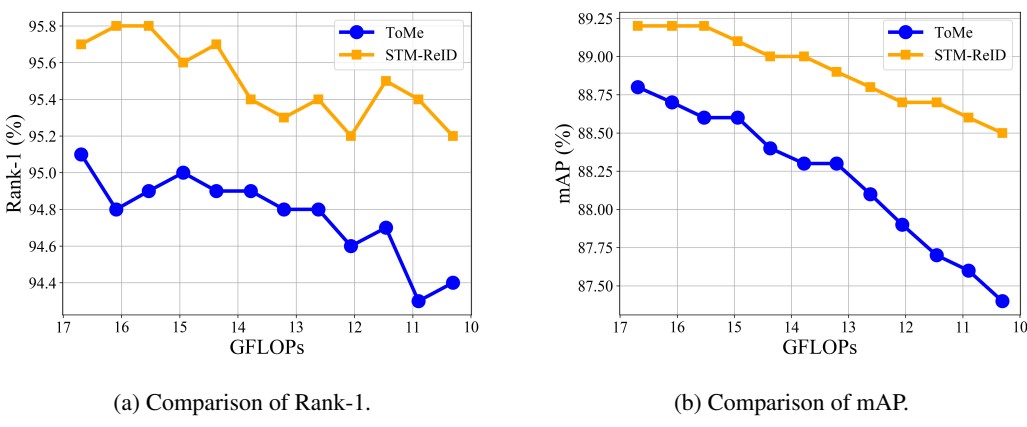

(a) Comparison of Rank-1.

(b) Comparison of mAP.

Figure 6: Comparison with ToMe under varying GFLOPs on Market1501 dataset.

performance superiority across different computational budgets, attributable to our method's refined perception of critical local regions.

## D  VISUALIZATION

### D.1  VISUALIZATION OF L2 NORM DISTRIBUTIONS UNDER DIFFERENT STRATEGIES

Taking the fourth layer of the model as an example, Figure 7 illustrates the $L2$ norm distributions of features using different strategies on Market1501. "Original" refers to the norm distribution of raw features before fusion, shown in blue. "Strategy1" corresponds to the result of traditional token size-based weighted averaging, and "Strategy2" corresponds to our PRN-Fuse. Both Strategy1 and Strategy2 are shown in light red in both subfigures (areas overlaid on the blue background of the original distribution appear dark red). "Density" here refers to the probability density of feature norms, quantifying the proportion of samples with different norm values. High-norm regions (right side of each subfigure) usually correspond to highly discriminative key features (e.g., clothing pat-

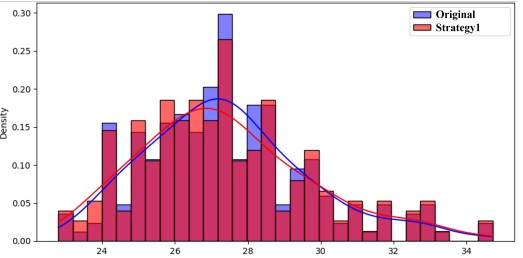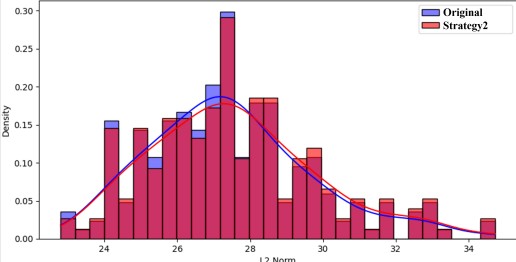

Figure 7: The $L2$ norm distributions of features before and after fusion using different strategies on Market1501. Strategy 1 denotes token fusion through token size-based weighted averaging, while strategy 2 denotes our PR-Degree weighted token fusion with norm interpolation.

terns, limb details), while low-norm regions (left side) mostly represent background or redundant information. Notably, compared with the original norm distribution, $Strategy1$ exhibits marked contraction in high-norm regions post-fusion. The global downward shift of feature norms indicates energy dissipation in original features, potentially compromising identity discriminability. The distribution shows that the density of high-norm regions in Strategy1 decreases significantly, indicating that traditional fusion causes norm diminishment of key features; in contrast, the distribution of Strategy2 is closer to "Original", especially with high-norm regions preserved, confirming the effectiveness of PRN-Fuse in alleviating norm diminishment.

## D.2    VISUALIZATION OF TOKEN MERGING RESULTS ON MARKET1501

We conduct visualization experiments demonstrating the compression effects across the first 10 layers, as shown in Figure 8. The number of merged tokens progressively increases as the layers deepen. Notably, our strategy predominantly merges tokens in background regions while effectively preserving foreground critical information.

## D.3    VISUALIZATION OF RETRIEVAL RESULTS ON MARKET1501

To provide an intuitive evaluation of our method, we conduct visualization experiments comparing ToMe and STM-ReID on the Market1501 dataset, as illustrated in Figure 9. Specifically, given a query image, we retrieve the top-5 gallery images ranked by similarity scores. The visual results demonstrate that our STM-ReID exhibits superior robustness compared to ToMe across diverse scenarios. With the proposed token merging mechanism, STM-ReID effectively focuses on critical local regions and achieves good retrieval results.

## D.4    VISUALIZATION OF FEATURE DISTRIBUTIONS

We perform t-SNE visualization to analyze feature distributions, as shown in Figure 10. From left to right are TransReID, ToMe, and our STM-ReID, respectively. The regions highlighted by red and black bounding boxes reveal two critical limitations of ToMe: 1) features of the same identity exhibit substantial intra-class distances; 2) features from different identities become erroneously entangled. It leads to degraded matching performance during inference. In contrast, our STM-ReID achieves significantly reduced intra-class distances and well-separated cluster-like distributions across identities. These results substantiate the superiority of our token merging strategy, which stems from its focused attention on critical local regions.

# E    LLM USAGE

Large Language Models (LLMs) aided this manuscript's writing and polishing, focusing solely on linguistic improvements—such as refining language, enhancing readability, and rephrasing sentences. The LLM was not involved in ideation, methodology, or experimental design; all scientific content and analyses are the authors' work. The authors take full responsibility for the manuscript, including LLM-polished text, which adheres to ethical guidelines.

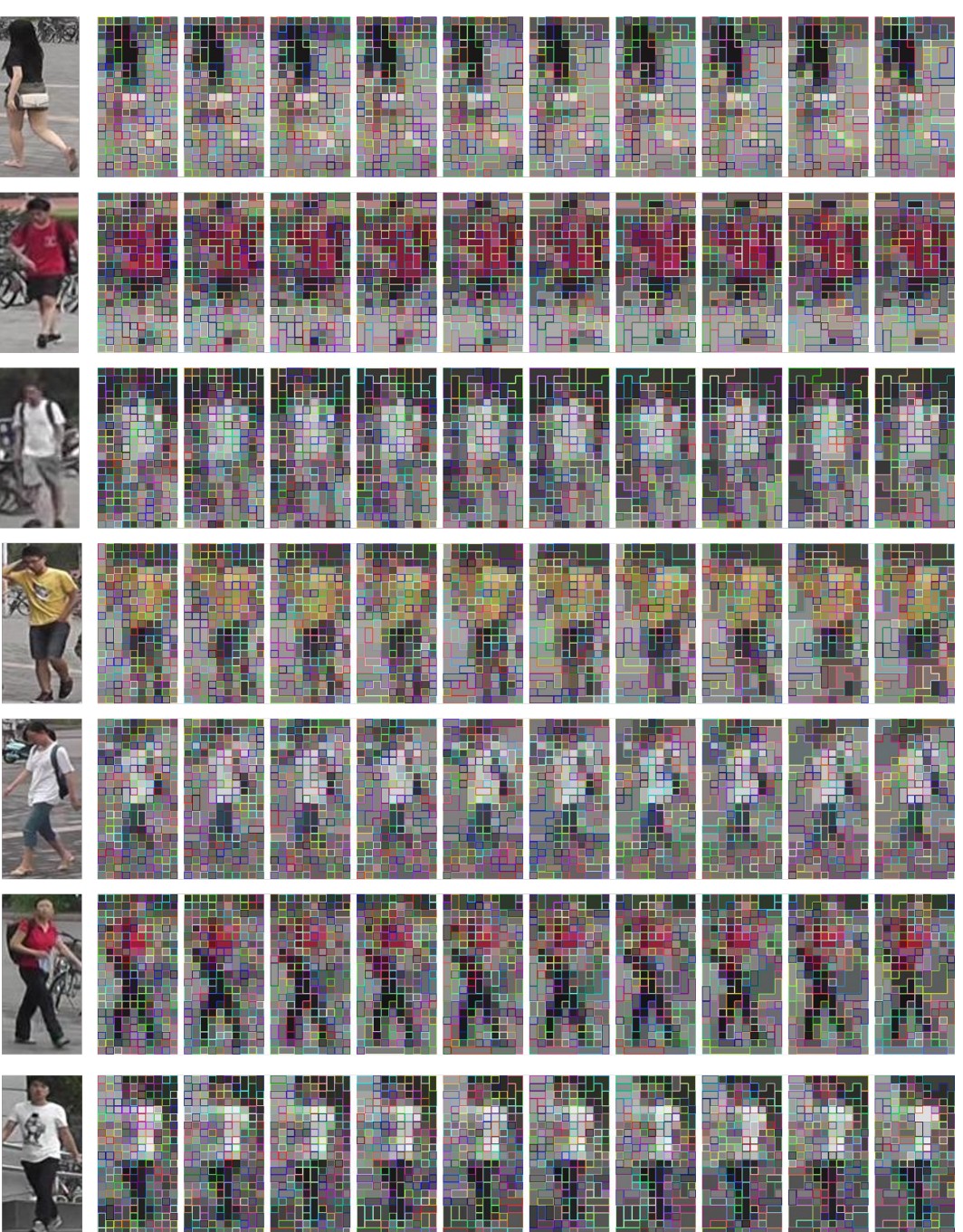

Figure 8: Visualization of token merging results.

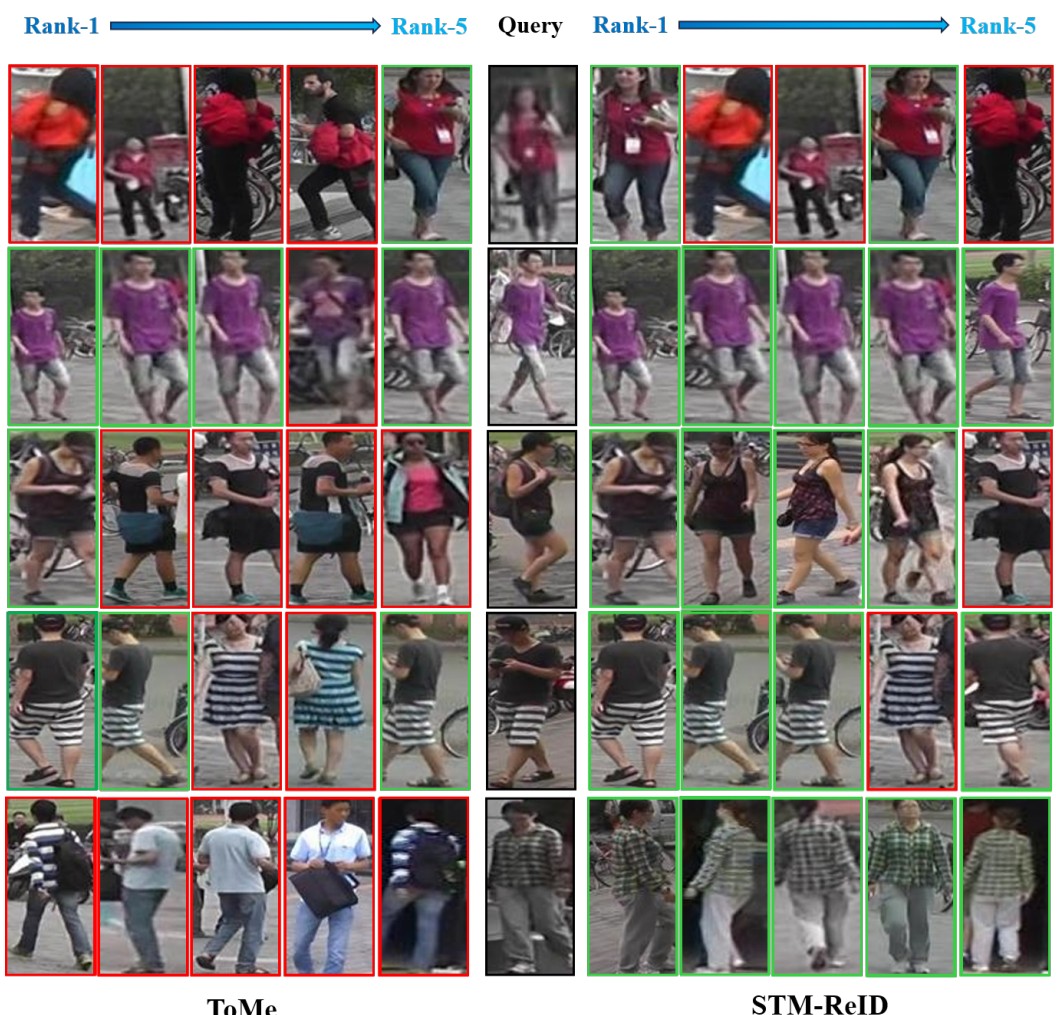

Figure 9: Visualization of retrieval results.

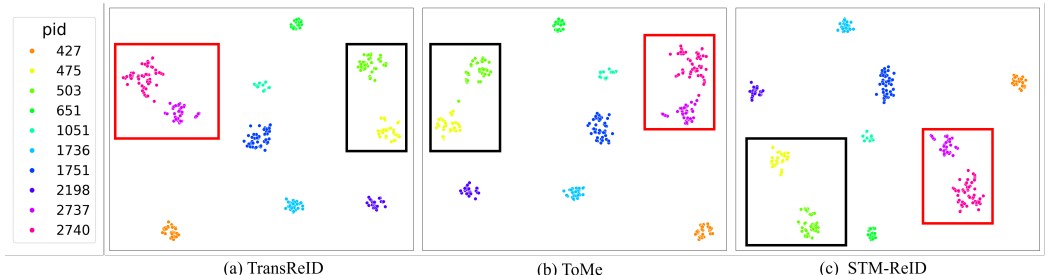

Figure 10: Visualization of 10 randomly selected identities from the MSMT17 dataset via t-SNE, where different identities are indicated in different colors.

