# OpenReview forum: "Lighter is Better: Boost Your ViT in Person Re-Identification via Spatial-Aware Token Merging"
_ICLR.cc/2026/Conference — ICLR 2026 Conference Withdrawn Submission_

### Official Review · Reviewer_ypfN · 2025-10-27

**Soundness:** 2
**Presentation:** 3
**Contribution:** 2
**Rating:** 4
**Confidence:** 3

**Summary:**

Addressing two major pain points in existing lightweight ReID methods：(1) token pruning tends to discard discriminative contextual information, and (2) existing token merging methods target image classification and overlook the local cues required for ReID. This paper proposes the STM-ReID framework. It is the first to infuse spatial awareness into the entire token merging pipeline. Being training-free (which reduces deployment costs), it is specifically tailored for ViT-based ReID, achieving the goal of "reducing computational cost while preserving identity-relevant local details."

**Strengths:**

The paper's strengths are rooted in its appealing practical value. It identifies a relevant problem：ViT's inefficiency in ReID，and delivers a training-free solution that is both effective and readily deployable. By tailoring the token merging process to ReID, it outperforms pruning and avoids the need for fine-tuning. The significant reductions in GFLOPs and gains in inference speed, as validated by experiments, underscore its potential value for real-world scenarios like edge computing and real-time surveillance.

**Weaknesses:**

1. The paper's contributions are reasonable, but the experimental validation is insufficient. The work is inspired by [1] (Feng & Zhang, 2023), adopting the core idea of reducing complexity by merging tokens rather than pruning. Its primary novelty lies in transferring and adapting this concept from image classification to the ReID domain, which constitutes an application-driven contribution rather than a fundamentally original innovation.
2.  Another key contribution is its training-free nature. This is a clear and practical idea, significantly lowering the deployment barrier. However, a inherent limitation of being training-free is the potential compromise in generalization and robustness. The method heavily relies on the strong spatial prior that "pedestrians are typically centered," an assumption often violated in real-world scenarios which are far more complex. Crucially, the paper does not evaluate its method on images with off-center compositions, occlusions, or geometric transformations (e.g., rotation, cropping). Therefore, its generalization capability to complex, real-world environments remains highly questionable.
3. As the author did not open the source code of the paper, I am unable to verify its experimental results. I suggestion the author to open the code and experimental data on platforms such as Github, and provide a detailed explanation of the experimental setup to promote the community.
[1]  Zhanzhou Feng and Shiliang Zhang. Efficient vision transformer via token merger. IEEE Transactions on Image Processing, 32:4156–4169, 2023.

**Questions:**

1.Spatial Prior and Robustness: The core of your method relies heavily on the spatial prior that pedestrians are centered. However, this assumption is frequently violated in real-world scenarios, such as when pedestrians are at the image edge, or under wide-angle lens distortion or heavy occlusion. How does STM-ReID perform on images with off-center composition or significant occlusion? Do you have any data or plans to assess potential performance degradation in these challenging conditions? To better demonstrate generalization, we suggest including results on more complex ReID datasets like Occluded-DukeMTMC or Partial-REID in the rebuttal, with specific analysis on their "off-center" and "occluded" samples. This would provide the most direct evidence for the robustness of your approach.
2.Sensitivity to Transformations: Your method explicitly uses Dynamic Virtual Coordinates to encode absolute spatial locations. Could this make the model particularly sensitive to geometric transformations like rotation, translation, or cropping? Since these transformations alter token coordinates without changing the person's identity, have you evaluated the model's robustness in such cases? A comparative experiment—applying simple transformations like ±15° random rotations to the test set and observing the performance change of STM-ReID versus the base model—could strongly indicate whether your method learns genuinely position-invariant identity features.
3.The Trade-off of Being Training-Free: The training-free design is a notable innovation for ease of deployment. However, as it does not update the model's underlying features, could a minimal fine-tuning stage after merging potentially lead to higher accuracy? To clarify this trade-off, could you provide experimental results under cross-domain settings or controlled perturbations? This would help the community better understand the relative performance and generalization of your method compared to fine-tuning-based alternatives.
4.Qualitative Evidence:To more intuitively demonstrate that crucial local details are preserved after token merging, it would be very insightful to visualize and compare the attention heatmaps (e.g., from the [CLS] token) before and after the merging process. Showing that the model's focus remains on discriminative regions post-merging would offer strong qualitative support for your claims.

**Details Of Ethics Concerns:**

No ethical concerns were identified. The content complies with relevant standards, with no issues regarding privacy, fairness, safety, or rights. No further ethics information is needed.

---

### Official Review · Reviewer_eRZZ · 2025-10-28

**Soundness:** 3
**Presentation:** 3
**Contribution:** 2
**Rating:** 4
**Confidence:** 5

**Summary:**

This paper proposes STM-ReID, a lightweight pedestrian re-identification (ReID) method based on Vision Transformer (ViT), which employs a token merging strategy to reduce computational overhead.

**Strengths:**

1. The idea and framework is straightforward and intuitive.
2. The authors conducted extensive experiments on representative datasets spanning four ReID domains.

**Weaknesses:**

1. Combining token merging with ReID holds limited significance. While it may offer potential benefits for large-scale models such as GPT-5, DeepSeek-R1, or those exceeding one billion parameters, it remains largely irrelevant to most ReID systems that employ ViT-Base as the backbone. In practical ReID applications, inference speed is primarily constrained by feature retrieval, for which specialized Hash-ReID methods already exist. Consequently, feature extraction is not the primary performance bottleneck.

2. Although the paper qualitatively claims that token merging “better preserves spatial and contextual information,” this assertion requires explicit quantitative evidence to substantiate its advantages over pruning strategies in maintaining contextual continuity. For instance, it remains unclear whether the reported performance improvements are statistically significant under challenging conditions, such as occlusion or complex background scenes. Furthermore, robustness validation on the Occluded-Duke dataset is absent.

**Questions:**

Can the authors validate the significance of the inference speed improvement in ViT-Large or larger models?

---

### Official Review · Reviewer_ru3s · 2025-10-30

**Soundness:** 2
**Presentation:** 3
**Contribution:** 2
**Rating:** 4
**Confidence:** 4

**Summary:**

This paper addresses the problem of high computational cost in ViT-based person ReID systems, which is a barrier to real-time deployment. It critiques existing lightweight methods, those like token pruning (which loses discriminative context) and generic token merging (designed for classification and overlooks local cues specific to ReID), and proposes STM-ReID, a training-free, spatially-aware token merging framework for ViT-based ReID.

**Strengths:**

1. Avoiding additional training/finetuning is a pragmatic choice as it reduces the computational overhead for end users.


2. Unlike many lightweight ReID papers that only report GFLOPs, STM-ReID also reports inference time and throughput (Table 1), providing a more comprehensive view of practical real-world efficiency, which is critical for the application of ReID in surveillance.


3. Table 5 systematically tests the impact of DSE-Assess, CCF-Match, and PRN-Fuse, demonstrating that each component contributes to performance improvement.

**Weaknesses:**

1. The overall framework of the paper uses ToMe [2] as the baseline for ReID performance, but the proposed core components lack true innovation and are merely incremental improvements. Specifically as follows:

    (1) DSE-Assess: The concept of "spatial-aware entropy" is derived from EViT [1], which uses attention entropy to identify redundant tokens. STM-ReID only adds a Gaussian spatial prior, which is a minor modification rather than a paradigm shift.

    (2) CCF-Match: The bipartite soft matching is directly borrowed from ToMe [2], with the only change being the addition of the responsiveness term (PR-Degree) — a simple multiplicative factor. No new matching logic is introduced.

    (3) PRN-Fuse: Norm-preserving fusion using max-norm interpolation is a standard technique in feature fusion (e.g., FusionReID [3]). Applying it to token merging does not constitute original work.

    [1] EViT: Expediting Vision Transformers via Token Reorganizations[C]//ICLR 2022.

    [2] Token Merging: Your ViT But Faster[C]/ICLR 2023.

    [3] Unity is strength: Unifying convolutional and transformer features for better person re-identification[J]. IEEE Transactions on Intelligent Transportation Systems, 2025.

2. ReID systems often handle occluded pedestrians (e.g., people blocked by trees), but STM-ReID is only tested on full-body datasets (Market1501, MSMT17). It is not tested on occlusion-focused ReID datasets such as Partial-ReID, so no evidence supports the claim of "preserving local details." Occlusion can disrupt the spatial neighborhood information DSE-Assess depends on. Moreover, on the already tested datasets, it offers no significant performance improvement over ToMe and requires longer inference time.

3. The paper mentions that STM-ReID is slightly slower than ToMe (46.28s vs. 45.99s, Table 1), but fails to provide a breakdown of which component contributes to the latency — such as whether the entropy calculation of DSE-Assess or the bipartite matching of CCF-Match contributes more to the overhead. Without this information, readers are unable to determine how to optimize STM-ReID for faster inference.

4. Token merging inherently diminishes token diversity, which is critical for fine-grained matching in ReID. The paper does not discuss how STM-ReID mitigates this loss of diversity — e.g., does merging 8 tokens per layer erase subtle differences between clothing patterns? This oversight overlooks the core trade-off inherent to token merging.

**Questions:**

1. The neighborhood size (k=9) is selected based on Market1501. Have you tested different values of k (i.e., 4, 9, 16) on MSMT17 or other datasets? If k needs to be adjusted for each dataset, does STM-ReID maintain practical feasibility for real-world deployment, where dataset characteristics vary?

2. STM-ReID relies on spatial neighborhoods to preserve local details. How does it perform with occluded pedestrians, such as those in Partial-ReID?

3. Which component (DSE-Assess, CCF-Match, PRN-Fuse) accounts for the largest proportion of the latency difference compared to ToMe? Can you provide a latency breakdown for each component to guide optimization?

---

### Official Review · Reviewer_eWLK · 2025-10-31

**Soundness:** 2
**Presentation:** 2
**Contribution:** 2
**Rating:** 4
**Confidence:** 4

**Summary:**

This paper proposes STM-ReID, a novel training-free, spatial-aware token merging framework for accelerating Vision Transformer (ViT)-based person re-identification (ReID) without sacrificing accuracy. Instead of relying on token pruning, which risks discarding crucial local details, STM-ReID introduces a three-stage merging pipeline that includes: DSE-Assess: A dynamic, entropy-based token importance assessment that considers spatial priors and head-wise diversity; CCF-Match: A correlation-guided bipartite soft matching strategy that avoids merging semantically critical tokens;	PRN-Fuse: A norm-preserving fusion method to mitigate feature attenuation during token merging. Experiments on Market1501 and MSMT17 datasets demonstrate that STM-ReID achieves up to 24% reduction in GFLOPs and competitive or even better accuracy compared to both ReID-specific and generic token compression methods. The framework also generalizes well to classification tasks on ImageNet-1k. Ablation studies and visualizations validate the contribution of each component.

**Strengths:**

- No need for retraining or fine-tuning—STM-ReID operates solely at inference using attention maps and token features from pretrained models.
- Strong empirical results. Ablation studies, visualizations (token maps, retrieval, t-SNE), and algorithm details support the method’s effectiveness.
- Matches or outperforms methods like ToMe in both GFLOPs and accuracy.

**Weaknesses:**

- Limited Novelty in Problem Formulation: While the merging mechanisms are tailored to ReID, the overall idea of token merging is based on prior work like ToMe. The paper’s novelty is more in engineering refinement than in core methodological innovation.
- “Training-Free” Claim May Be Overstated: The method requires significant architectural modifications and additional computation at inference (e.g., PR-Degree, entropy-based head weighting), which may increase latency. No real-world inference time benchmarks on edge devices are provided to support deployment claims.
- Limited Analysis of Failure Cases: The paper lacks discussion or visualization of when and why STM-ReID may fail (e.g., highly occluded identities, cluttered scenes).
- Not End-to-End: The proposed framework cannot be trained end-to-end and depends entirely on precomputed attention maps, which may limit its adaptability in joint training setups (e.g., joint optimization of token importance with backbone features).

**Questions:**

See weaknesses

---

### Note · Authors · 2026-01-14

I have read and agree with the venue's withdrawal policy on behalf of myself and my co-authors.